# Short-Term Transcriptomic Points of Departure Are Consistent with Chronic Points of Departure for Three Organophosphate Pesticides across Mouse and Fathead Minnow

**DOI:** 10.3390/toxics11100820

**Published:** 2023-09-29

**Authors:** Rubia Martin, Monique Hazemi, Kevin Flynn, Daniel Villeneuve, Leah Wehmas

**Affiliations:** 1Office of Research and Development, Center for Computational Toxicology and Exposure, Chemical Characterization and Exposure Division, Oak Ridge Institute for Science and Education, U.S. Environmental Protection Agency, Durham, NC 27709, USA; martin.rubia@epa.gov; 2Office of Research and Development, Center for Computational Toxicology and Exposure, Great Lakes Ecology Division, Oak Ridge Institute for Science and Education, U.S. Environmental Protection Agency, Duluth, MN 55804, USA; hazemi.monique@epa.gov; 3Office of Research and Development, Center for Computational Toxicology and Exposure, Great Lakes Ecology Division, U.S. Environmental Protection Agency, Duluth, MN 55804, USA; flynn.kevin@epa.gov (K.F.); villeneuve.dan@epa.gov (D.V.); 4Office of Research and Development, Center for Computational Toxicology and Exposure, Chemical Characterization and Exposure Division, U.S. Environmental Protection Agency, Durham, NC 27709, USA

**Keywords:** benchmark dose, fathead minnow, mouse, organophosphate pesticides, transcriptional points of departure

## Abstract

New approach methods (NAMs) can reduce the need for chronic animal studies. Here, we apply benchmark dose (concentration) (BMD(C))–response modeling to transcriptomic changes in the liver of mice and in fathead minnow larvae after short-term exposures (7 days and 1 day, respectively) to several dose/concentrations of three organophosphate pesticides (OPPs): fenthion, methidathion, and parathion. The mouse liver transcriptional points of departure (TPODs) for fenthion, methidathion, and parathion were 0.009, 0.093, and 0.046 mg/Kg-bw/day, while the fathead minnow larva TPODs were 0.007, 0.115, and 0.046 mg/L, respectively. The TPODs were consistent across both species and reflected the relative potencies from traditional chronic toxicity studies with fenthion identified as the most potent. Moreover, the mouse liver TPODs were more sensitive than or within a 10-fold difference from the chronic apical points of departure (APODs) for mammals, while the fathead minnow larva TPODs were within an 18-fold difference from the chronic APODs for fish species. Short-term exposure to OPPs significantly impacted acetylcholinesterase mRNA abundance (FDR *p*-value <0.05, |fold change| ≥2) and canonical pathways (IPA, *p*-value <0.05) associated with organism death and neurological/immune dysfunctions, indicating the conservation of key events related to OPP toxicity. Together, these results build confidence in using short-term, molecular-based assays for the characterization of chemical toxicity and risk, thereby reducing reliance on chronic animal studies.

## 1. Introduction

The identification of points of departure (PODs) is an essential step in chemical risk characterization. PODs have generally been derived from acute-, subchronic-, or chronic-toxicity dose–response animal studies via observations of apical endpoints such as mortality, body/organ weight change, disease formation, altered function/behavior, etc. However, the studies traditionally used to derive PODs are time intensive, costly, and rely heavily on the use of large numbers of animals, presenting ethical concerns. In the past, a common and reliable approach to set a POD for a chemical was to identify no- and/or lowest-observed-adverse-effect levels (NOAELs, LOAELs) from chronic animal studies and apply risk assessment principles and safety factors to estimate permissible exposure/intake values. More recently, benchmark dose (BMD) analysis [1] has replaced the NOAEL/LOAEL approach as the preferred method for identifying PODs in chemical risk assessment [2]. BMD analysis fits mathematical models to experimental data points to interpolate a BMD (defined as the dose required to significantly increase the probability of extra risk) and the lower statistical bound of the BMD (i.e., BMDL) generally sets the POD. This method is considered superior for identifying a POD because it does not solely depend upon the study design and uses data from the entire dose response curve. Despite this significant advance in chemical risk assessment, reliance on chronic animal studies for identifying PODs of individual chemicals cannot keep pace with the large number of chemicals in commerce and the environment. Thus, new approach methodologies (NAMs) that can replace, reduce, and/or refine (3Rs) animal usage, while improving throughput and efficiency of chemical risk assessment strategies, are desired [3].

The U.S. Environmental Protection Agency’s (U.S. EPA) Office of Pesticide Programs has the authority under the Federal Insecticide Fungicide and Rodenticide Act (FIFRA) [4] to require toxicity data from registrants to support their risk assessments. The use of the adverse outcome pathway (AOP) framework in conjunction with higher throughput NAMs can provide meaningful biological context and build confidence in their use for risk assessment, helping to reduce testing requirements and facilitating the transition towards the use of more prospective molecular endpoints [5,6]. The combination of molecular-based transcriptomic measurements and BMD analysis has gained a lot of attention for its potential to provide PODs from short-term exposures that are typically within the 10-fold range of those obtained from traditional, longer-duration toxicity studies [7,8,9,10]. Moreover, these comparisons have shown correlation with other apical studies at several time points, including acute-, subchronic-, and chronic-toxicity studies [8]. Many of these proof-of-principle studies have been efficacious in rodents for human health applications with fewer demonstrating efficacy for ecological health. In the context of the AOP framework, these studies suggest that upon chemical exposure, gene expression changes occur in a dose-dependent manner, which generally precede the adverse effects of toxicological concern. If we perform the BMD modeling of these early transcriptomic changes, we can identify a POD below which no concerted molecular changes are expected to occur, and therefore, no apical effects are expected either [7]. Thus, short-term transcriptomic PODs (TPODs) may be more conservative than traditional apical PODs (APODs) while helping to advance the pace of chemical risk assessments, which could also help to reduce the number of pesticide tests required by registrants through toxicity testing waivers or other means.

To further explore and evaluate these ideas, we aimed to provide two comparative case studies, demonstrating current applications of state-of-the-art genomic technologies combined with BMD modeling to better understand the molecular effects and potency of three well-studied organophosphates pesticide (OPPs): fenthion (FT), methidathion (MT) and parathion (PT). We hypothesized that changes in gene expression from short-term in vivo exposures can be modeled to identify a BMD and POD that is consistent with the traditional adverse effects of chronic exposures, using a model species of human health (mouse) and a model species of ecological health (fathead minnow). Although FT, MT, and PT are no longer used or produced in the U.S., they are still used as pesticides in other parts of the world [11,12] and ubiquitously detected in the environment [13], posing potential risks to aquatic organisms and human health. Exposure to OPPs is known to cause neurotoxicity, teratogenicity, endocrine modulation, compromised cognitive development, and damage to reproductive and immune systems [14]. OPPs have a wealth of existing chronic toxicity data readily available and a well-established mode of action (i.e., acetylcholinesterase inhibition), making them ideal candidates for the present case studies. Our results provide human and ecological health-focused evidence of an approach that could potentially conserve economic resources and waive time-intensive chronic animal testing requirements.

## 2. Materials and Methods

### 2.1. Chemicals

Mouse liver study: OPP standards (fenthion (FT), methidathion (MT), and parathion (PT)) were obtained from Chem Service Incorporation (West Chester, PA) (Table 1) and incorporated into AIN-93G rodent diet (or not for controls) by TestDiet (Richmond, IN, USA).

Fathead minnow larva study: OPP standards (FT, MT, and PT) were procured from Sigma-Aldrich (St. Louis, MO, USA) (Table 1), and exposure solutions were prepared by Great Lakes Toxicology and Ecology Division, Duluth, MN, USA. 

### 2.2. Mouse Liver Study Design

All methods involving the mouse exposure study design were previously described by Rooney and colleagues who were investigating transcriptomic signatures of hepatocarcinogenic and non-hepatocarcinogenic OPPs [15]. For the present work, liver samples were repurposed to test our hypothesis; therefore, we were limited to tissues and in-life measurements available from that original study [15]. Briefly, three-week-old B6C3F1 male mice were purchased from Charles River Laboratories and maintained at the AAALAC-accredited U.S. EPA animal facility (Research Triangle Park, NC, USA) under conditions recommended by the National Research Council Guide for the Care and Use of Laboratory Animals [16]. All animal use and procedures were approved by the Institutional Animal Care and Use Committee of the U.S. EPA National Health and Environmental Effects Research Laboratory. Mice were housed in solid-bottom cages with Alpha-dri^®^ bedding (Sheppard Specialty Papers) at *n* = 4/cage for social welfare considerations, while Nestlets (Anacare) and Nylabones were provided for enrichment. The temperature was maintained at 21 ± 1 °C, humidity at 50 ± 10%, and lighting on a standard 12 h light:12 h dark cycle. Water and food were freely accessible with consumption monitored [15].

Following acclimation, cages were randomly assigned to the following treatment groups: MT at 10, 25, 50, 100 ppm, FT at 2.5, 5, 10, 20 ppm, PT at 20, 40, 80, 160 ppm (*n* = 16/dose/group), or control groups, stratified by average body weight per cage. However, due to high levels of morbidity (30%) and mortality (40%), the 160 ppm PT exposure group was excluded from the present study [15]. These OPP levels correspond to estimated intake levels of 1.5, 3.75, 7.5 and 15 mg/Kg-bw/day for MT; 0.375, 0.75, 1.5 and 3 mg/Kg-bw/day for FT; and 3, 6, 12 and 24 mg/Kg-bw/day for PT, respectively (Figure 1). Exposure to OPP treatment or control diet began at eight weeks of age. Dose selections were based on previous chronic studies [17,18,19]. Body weight was monitored throughout this study.

Following seven days of exposure to the OPPs, changes in body weight, relative liver weight, serum liver enzyme chemistry, hepatic cytochrome P450 activity (*Cyp1a*, *Cyp2b*, and *Cyp3a*), liver histopathology, and liver cell proliferation were measured and reported by Rooney and colleagues [15]. To summarize, no significant effects on serum liver enzymes, liver histopathology, or liver proliferation were observed after seven days of exposure. Only PT caused significant reductions in body and relative liver weight at 40 and 80 ppm, whereas only FT caused a significant, intermittent increase (no dose-dependent trends) in hepatic cytochrome P450 enzyme activity at either 2.5 or 5 ppm across the three assays [15]. 

### 2.3. Fathead Minnow Larva Study Design

All larval fathead minnows used for testing were provided by an on-site aquatic culture facility at the U.S. EPA’s Great Lakes Toxicology and Ecology Division, Duluth, MN. Briefly, fathead minnow embryos were collected from multiple crosses of at least three females in paired breeding. Fertilized embryos that were attached to polyvinyl chloride breeding tiles were transferred to a bath of aerated Lake Superior water and held at 25 °C. At five day after fertilization, larvae that had hatched and were freely swimming were collected and randomly sorted for use in the assay. Exposures to FT, PT, and MT were conducted in triplicate 96-well plates for 24 h (catalog #502162, NEST Scientific USA, Woodbridge, NJ, USA). Larvae were transferred, individually in 102 ± 3 µL Lake Superior Water, to each well, using a 0.5 cm glass capillary tube connected to a 200 µL manual pipettor via a short length of rubber tubing. An additional 600 µL of exposure media was added directly to each well. Each plate included 12 different treatments with eight biological replicates (individual larval fish) per treatment. There were 10 nominal concentration dilutions of the test chemical (PT: 0.00011—35 mg/L; MT: 0.0103—103 µg/L; or FT: 0.00021–6.6 mg/L) plus a control (Lake Superior water) and a CuSO_4_ positive control (150 µg/L). All exposure wells, including control and positive control wells, contained 0.25% dimethyl sulfoxide. A dissecting microscope was used to verify that each well contained 1 submerged viable organism. Plates were then sealed with flexible silicone mats (catalog #506065, NEST Scientific USA, NJ, USA) to limit evaporation and headspace. Plates were incubated at 25 °C on a 16 h:8 h light/dark cycle (myTemp Mini, Benchmark Scientific, Sayreville, NJ, USA). After the 24 h exposure, survival and/or any abnormalities were determined via observation of each individual larva, using a dissecting microscope. 

Post-exposure OPP concentrations in the test wells were evaluated by collecting 250 µL of the exposure water and adding it to 250 µL acetonitrile, and then storing frozen at −20 °C. Three replicate exposure water samples per treatment were selected at random for analysis. Then, media samples were analyzed for chemical concentration levels, using liquid chromatography mass spectrometry (LC-MS). Only the measured concentrations levels were considered for analysis (Figure 1). A complete explanation of LC-MS methodological details is included in the Appendix A along with a summary of the measured concentrations for each OPP.

### 2.4. RNA Extraction and Isolation

Mouse liver study: After 7 days of exposure, mice were euthanized via asphyxiation using carbon dioxide followed by cardiac puncture. We speculated that the liver, a major metabolic nexus of toxicant insult, might serve as a valid surrogate for measuring organophosphate toxicity at the transcriptomic level. Thus, a subset of seven liver samples per each treatment condition were selected for global gene expression analysis. During tissue harvest, livers were weighed; left lateral, caudate and right medial liver lobes were trimmed; and portions of each lobe were snap-frozen in liquid nitrogen. Frozen samples were stored at −80 °C. RNA was isolated and homogenized from ~20 mg of archived frozen liver tissue, using RNAzol^®^RT (Molecular Research Center, Cincinnati, OH, USA). Then, RNA was purified using the RNeasy MinElute column protocol (Qiagen GmbH, Hilden, Germany) and a subset of samples evaluated for RNA integrity (RIN ≥ 6.6), using the Agilent 2100 Bioanalyzer RNA Nano Chip (Agilent Technologies GmbH, Berlin, Germany). RNA was quantified spectrophotometrically (NanoDrop Technologies, Wilmington, DE, USA). RNA samples were isolated and purified at Expression Analysis Genomic Services, Q2 Solutions (Durham, NC, USA), or at the U.S. EPA (Durham, NC, USA).

Fathead minnow larva study: Whole larvae were homogenized in Buffer RLT + beta-mercaptoethanol (Qiagen RNeasy) before transferring to 96-well plates to proceed with RNA isolation using MagMAX-96 Total RNA isolation kits (Life Technologies Corporation, AM1830, Carlsbad, CA, USA) following manufacturer protocol. RNA was quantified using a Take3 microvolume plate (Agilent, Santa Clara, CA, USA) and a Synergy HTX plate reader (BioTek, Winooski, VT, USA). Quality of RNA isolation was further confirmed by testing a random set of 16 samples per 96 well plate, using an Agilent Tapestation (Agilent, Santa Clara, CA, USA). RNA was isolated from eight individual larvae per treatment, excluding any individuals that had died during the exposure. An additional 27 samples were not sequenced due to insufficient RNA, poor RNA quality, or failed library preparations (RIN < 9). Total RNA from individual fish (250 ng/sample) was then used to prepare sequencing libraries with SENSE mRNA-Seq Library Prep kits (Lexogen GmbH, Vienna, Austria), with 15 cycles of PCR amplification. The quality and size of the libraries were checked on an Agilent TapeStation system, and each pooled library for sequencing contained the full complement of samples (excluding mortalities and failed samples) from the exposure plate (up to 96 samples). 

### 2.5. RNA Sequencing

Mouse liver study: Changes in gene expression were measured on purified RNA by Templated Oligo-Sequencing (TempO-Seq, BioSpyder Technologies, Carlsbad, CA, USA) using the mouse whole transcriptome assay (30,146 probes targeting 21,448 genes) as previously described by Yeakley et al. 2017 [20]. Predesigned detector oligo (DO) pairs were bound near each other on mouse mRNA target sequence sites allowing ligation. Unbound DOs were removed, while bound and ligated DOs were amplified by PCR. During PCR, sequencing adaptors and sample specific barcodes compatible with Illumina sequencers and software were also introduced. The PCR-amplified and barcoded samples were pooled into libraries for 50 base-pair sequencing on an Illumina HiSeq2500. Sequenced reads were demultiplexed using Illumina software to give a FASTQ file for each sample. 

Fathead minnow larva study: RNA sequencing was conducted at the Michigan State University Research Technology Support Facility, Genomics Core, on a NovaSeq 6000 system employing a S4 2 × 150 bp flow cell (Illumina, San Diego CA, USA). Base calling was performed with Illumina Real Time Analysis v. 2.7.7 and demultiplexing was achieved using Illumina Bcl2fastq v.2.20.0. 

Raw FASTQ files and raw and normalized read count matrices for the mouse liver study were submitted to the National Center for Biotechnology Information (NCBI) Gene Expression Omnibus database (accession GSE240853). Raw FASTQ files for the fathead minnow larva study were submitted to the NCBI Sequence Read Archive (submission: SUB13852035).

### 2.6. Gene Expression Analysis

Both mouse liver and whole fathead minnow larva raw gene expression data were processed using the Partek Flow software pipeline (2022 Partek Inc., St. Louis, MO, USA). Briefly, read ends were trimmed by quality (Phred score <20) and length (<25 bp). Cleaned reads from the mouse liver study were aligned to *Mus musculus* (mm39) and cleaned reads from the fathead minnow larva study were aligned to the *Pimephales promelas* (EPA_FHM_2.0), using STAR v2.7.8a [21]. Aligned mouse liver reads were quantified to Ensembl transcript release 106, while aligned larval fathead minnow reads were quantified to Refseq GCA_016745375.1, using Partek E/M. For both studies, counts were filtered using a noise reduction function where it excludes any gene features with a median ≤ 8 counts across all samples. Counts were normalized to account for differences in library sizes, using counts per million mapped reads (CPM). A pseudocount of 1 was added to all genes prior to log_2_ transformation. Using a general linear model in Partek, mouse transcriptomics data were also adjusted for batch effects that resulted from experimental blocking and from the personnel who performed RNA isolations (Appendix A). No batch effect removal was necessary for the fathead minnow larva data. 

### 2.7. Transcriptomics Benchmark Dose (Concentration) Response Modeling and Analysis

Transcriptomic BMD (mouse liver study) and transcriptomic benchmark concentration (BMC, fathead minnow larva study) analysis was performed with BMDExpress 2.3 [22] and following an approach similar to the National Toxicology Program Approach to Genomic Dose–Response Modeling Report [10]. Briefly, the filtered, normalized, transformed, and batch-adjusted count matrix for the mouse liver study was mapped using mm10 for all doses calculated as mg/Kg-bw/day. Because fathead minnow is not represented in the BMDExpress genome reference libraries, and in order to access downstream GO enrichment and molecular pathway analyses, we translated all the identified fathead minnow genes to zebrafish homologs, using previous published interspecies homologous conversion data via basic local alignment search tool (BLAST) [23]. Thus, for the fathead minnow larva study, the filtered, normalized, log_2_-transformed count matrix was mapped to the danRer11 Ensembl annotation before upload into BMDExpress 2.3 with exposure concentrations as mg/L. Once the count matrices were uploaded into the BMDExpress, data were prefiltered to identify genes/features likely to exhibit dose/concentration-dependent response, using one-way ANOVA (maximum absolute fold change ≥ 2, false discovery rate [24] (FDR)-adjusted *p*-value < 0.05 for the mouse liver study and maximum absolute fold change ≥ 2, *p*-value < 0.05 for the fathead minnow larva study). In the following analysis description, reference to BMD should be considered interchangeable with BMC. All genes demonstrating a significant treatment level effect were fitted to several statistical models implemented in BMDExpress 2.3 (Exp2, Exp3, Exp4, Exp5, Linear, Poly2, Poly3, Hill, and Power). Each model was run assuming constant variance and with a threshold benchmark response factor equal to 1.349 (10%) standard deviation. The best polynomial model was selected based on a chi-square test for nested models (*p*-value < 0.05), and the best overall model was selected based on the lowest Akaike information criterion. The Hill model was flagged if the ‘k’ parameter was < 1/3 the lowest positive dose. In the case where a flagged Hill model was identified as the best overall model fit, the next best model with a *p*-value > 0.05 was selected. This was used to select the BMD and upper and lower 95%-confidence limits of the BMD (BMDU and BMDL, respectively) for each gene. Genes with BMD values > the highest dose tested and BMDU:BMDL ratios > 40 were excluded from further analysis. Dose-responsive genes were then mapped to gene-ontology biological processes (GO: BP), using the GO: BP function in BMDExpress 2.3. Significantly enriched gene sets were defined by Fisher’s exact test two-tailed *p*-value < 0.05 and at least three-gene overlap with the gene set. The median BMD of the most sensitive, enriched GO: BP defined the transcriptomic BMD, and the transcriptomic BMDL defined the TPOD for each OPP and each species.

### 2.8. Apical Point of Departure Queries

To address our main hypothesis that changes in gene expression from short-term exposures can identify TPODs that are consistent with chronic APODs, we focused our comparisons on those endpoints. Specifically, data comparisons considered TPODs from our short exposures in the mouse and fathead minnow studies and APODs that were previously derived from subchronic/chronic exposures in several species of mammals and fish, respectively. Our main objective with these TPOD-to-APOD comparisons was to see how well a short-term study derived TPOD predicted the most sensitive chronic APOD. For rodents studies the APOD is typically derived from two-year exposures and for fish studies, the APOD is typically derived from 28-day or longer exposures depending upon the fish species tested. Here, we did not compare TPODs to the concurrently identified APODs because few if any significant apical effects occurred in the 7-day mouse liver study, which was briefly summarized in Section 2.2 and previously reported in [15], and no significant apical effects occurred in the 1-day fathead minnow larva study. Apical PODs for several model species were extracted from the CompTox Chemicals Dashboard V2.2 [25], which is integrated with and can access data from >30 databases, containing in vivo toxicity data for >50,000 chemical substances. Briefly, a batch search was performed for FT, MT, and PT extracting data from ToxVal [25] and ECOTOX knowledge databases [26]. Subchronic and chronic apical toxicity data for mammals (LOAELs and NOAELs in mg/Kg-bw/day) and fish (lowest- and no-observed-adverse-effect concentrations–LOAECs and NOAECs—in mg/L) were identified for comparisons with the short-term transcriptomic BMD(C)s and respective TPODs. No other attempts were made to match exposure duration and testing conditions or other design factors of these chronic apical studies.

### 2.9. Ingenuity Pathway Analysis

To investigate the underlying mechanisms of OPP-dependent changes in the gene expression profiles, in silico analyses with Ingenuity Pathway Analysis (IPA, Qiagen) were performed for all identified dose/concentration-responsive genes from BMDExpress. Two types of analysis were performed in IPA: (1) a canonical pathway analysis to compare the activation or inhibition of well-established signaling pathways and (2) a disease and biological network analysis to compare the regulation of genes in critical biological processes or functions. Significant enrichment was determined by *p*-values < 0.05. If available, predicted activation (positive values) or inhibition (negative values) of pathways was determined by z-score. Fathead minnow larva genes were mapped to mouse orthologs prior to IPA analysis using previous published interspecies homologous conversion data via basic local alignment search tool (BLAST) [23]. Enriched canonical pathways (*p*-values < 0.05) for each chemical and species were combined in an IPA comparison analysis, which ranked pathways by absolute z-score across all groups.

## 3. Results

### 3.1. Short-Term Organophosphate Exposures Significantly Influenced Gene Expression 

Prefiltering of gene expression data identified 1099 (FT), 874 (MT), and 1039 (PT) differentially expressed genes (DEGs) from the mouse liver study following a 7-day exposure (ANOVA; FDR-adjusted *p*-value < 0.05; maximum absolute fold change ≥2 across dose levels) and 93 (FT), 295 (MT), and 152 (PT) DEGs from whole-body fathead minnow larvae following a 1-day exposure (Figure 2a) (ANOVA; *p*-value < 0.05; maximum absolute fold change ≥ 2 across dose levels). The comparison of overlapping DEGs across the OPPs (FT, MT, and PT) from the mouse liver study identified 445 that were shared. Such a substantial overlap suggests that these OPPs may be acting through a similar mode of action in the mouse liver. In contrast, in the fathead minnow larva study, only one shared gene between all three OPPs was observed (Figure 2b,c). Fewer DEGs, in general, were identified across the different OPPs for the fathead minnow larva study compared to the mouse liver study, which may be partially due to the differences in the study designs. The complete list of DEGs for both species can be found in Appendix A.

BMD(C) modeling identified 840 (FT), 689 (MT), and 805 (PT) dose-responsive DEGs in the mouse liver study and 52 (FT), 198 (MT), and 80 (PT) concentration-responsive DEGs in the fathead minnow larva study (Figure 2d). Over 76% of the ANOVA filtered genes for each OPP showed adequate BMD fits in the mouse liver study, whereas approximately 52% genes demonstrated adequate BMC fits in the fathead minnow larva study. When looking at the overlap of these dose/concentration-responsive DEGs, the mouse liver study identified a greater number of dose-responsive genes that were consistent across all OPPs (185) (Figure 2e), while the fathead minnow larva study had no overlap of concentration-responsive genes across all three OPPs. However, there were about one to ten concentration-responsive DEGs shared between any two OPPs (Figure 2f). Additionally, when reviewing the transcriptomic BMD(C) values for all modeled DEGs, log_10_-transformed median levels and ranges show relatively low variability and general consistency across the chemicals and species (Figure 2g) despite the differences in the study designs. Moreover, the differences in the relative potency of each OPP across the species are apparent at the gene level, suggesting that FT is the most potent OPP as it had the lowest median transcriptomic BMD(C) value in both species (Figure 2g). 

### 3.2. Gene Set-Based Transcriptomic Points of Departures Were Similar for Mice and Fathead Minnow Larvae

To establish gene set-based TPODs, we mapped all dose- and concentration-responsive DEGs to GO: BP categories. Enriched gene sets are typically preferred for TPOD identification, as they are believed to better reflect coordinated chemical-dependent biological responses as opposed to individual gene-level results [27]. Therefore, the median of the most sensitive enriched GO: BP established the BMD(C)s, while the associated lower bound defined the TPODs. These are summarized for each study and each OPP in Table 2. Both studies identified significant enrichment of GO: BP categories across all three OPPs. Specifically, the mouse liver study detected 742 (FT), 909 (MT), and 799 (PT) enriched gene sets, and the fathead minnow larva study detected 9 (FT), 48 (MT), and 109 (PT) enriched gene sets (Figure 3). Interestingly, the gene set BMD(C) and lower-bound accumulation plots or TPODs showed that FT shifted to the left, indicating FT as the most potent OPP in both species (Figure 3) similar to the gene-level results. Overall ranking for these OPP potencies identified FT > PT > MT in both species. When reviewing the gene set BMD(C) and TPOD values between both species, we found that they were surprisingly similar (within a fold change <1.5) given the differences in both studies (i.e., adult mouse liver vs. whole fathead minnow larvae; 7- vs. 1-day exposures; oral gavage vs. immersion; etc.). For instance, the gene set BMD (TPOD) values for FT, MT, and PT in the mouse liver study were 0.018 (0.009), 0.164 (0.093), and 0.101 (0.046) mg/Kg-bw/day, respectively, while the gene set BMC (TPOD) values for FT, MT, and PT in the fathead minnow larva study were 0.011 (0.007), 0.151 (0.115), and 0.090 (0.046) mg/L, respectively (Table 2).

### 3.3. Gene Set-Based Transcriptomic PODs Were Concordant with Chronic Apical PODs for Each Species

Ultimately, we aimed to compare the gene set-based TPODs derived from short-term exposure in mice and in fathead minnow to APODs derived from chronic exposures. Given the lack of BMD(C) data from traditional toxicity studies for FT, MT, and PT in mouse and fathead minnow, we used the chronic toxicity data consisting of chronic NOAEL(C)-based POD values from mouse and fathead minnow, when available, as well as multiple mammalian and fish species extracted from the CompTox Chemicals Dashboard. Notably, traditional apical BMD values are known to usually fall between the NOAELs and the LOAELs [28], making them a reasonable alternative for apical comparisons given the paucity of the BMD(C) data. For the mouse liver study, there were species-matched (mouse) chronic toxicity data reported for FT and MT with the most sensitive NOAELs based on the cholinesterase inhibition at < 0.03 and 1.6 mg/Kg-bw/day, respectively. The closest alternative for PT was the rat data with a chronic NOAEL of 0.1 mg/Kg-bw/day (Appendix A), which was also based on cholinesterase inhibition, making the chronic rodent APODs between 2.2- and 17-fold higher (less sensitive) than the gene set TPODs derived from the mouse liver study. For the fathead minnow larva study, there were also species-matched chronic data available for FT and MT with the most sensitive NOECs at 0.065 and 0.0063 mg/L, respectively. Chronic PT data were not available for fathead minnow, but the most sensitive NOEC for zebrafish was 0.0041 mg/L, resulting in the APODs that were 9.3-fold higher (less sensitive, FT) or between 11.2- and 18.3-fold lower (more sensitive, PT and MT, respectively) than the gene set TPODs derived from the fathead minnow larva study. Because the mouse or fathead minnow may not be the most sensitive species used in a human health or ecological assessment, respectively, we investigated how well these gene set TPODs performed as the general indicators of the most sensitive APODs from chronic studies conducted in several species of mammals (Figure 4a) or fish (Figure 4b). The details regarding the chronic apical studies used to identify the FT, MT, and PT APODs in mammals and in fish are presented in Appendix A, respectfully. The durations of these subchronic/chronic exposure-based studies ranged from 18 days to 2 years for mammals and from 28 days to 90 days for the fish species. When looking exclusively at the mammalian data, the most sensitive chronic NOAELs reported for FT, MT, and PT were < 0.02 (primate), 0.01 (dog), and < 0.01 mg/Kg-bw/day (dog), respectively (Appendix A). These values were within 2.2-, 9.3-, and 4.6-fold of the mouse-derived gene set TPODs for FT, MT, and PT, respectively. When looking at the fish species, the most sensitive NOECs were 0.0075 (rainbow trout), 0.0063 (fathead minnow), and 0.00019 mg/L (sheepshead minnow) for FT, MT, and PT, respectively (Appendix A). These values were within 1.1-, 18.3-, and 242.1-fold of the TPODs for FT, MT, and PT derived from the fathead minnow larvae, respectively. Interestingly, short-term exposures to these OPPs yielded gene set TPOD values that were generally within a 10-fold difference from the most sensitive subchronic/chronic-toxicity NOAEL(C)-based APODs for several species of mammals and fish (Figure 4a,b, respectively). Specifically, for mammals, the mouse liver study TPODs were generally more sensitive than and within a 10-fold difference from the APODs for 11 of 12 comparisons across all three OPPs. The average absolute fold difference (±s.d.) between the TPOD and APOD values for each OPP was 5.5 ± 2.6 for FT, 5.6 ± 7.7 for MT, and 2.7 ± 1.7 for PT. For fish, the fathead minnow larva study TPOD values were more sensitive than the two FT APOD comparisons. Across each of the OPPs, 2 of 6 TPOD-to-APOD comparisons were within 10-fold of each other with an average absolute fold difference (± s.d.) for FT and PT of 5.2 ± 2.8 and 85.5 ± 135.7, respectively. MT only fathead minnow APOD data for comparison. Interestingly, in both the mouse liver study and fathead minnow larva study, the TPODs for FT were lower than the APOD values representing a more conservative value. When comparing both the mouse and the fathead minnow TPODs to known NOAEL-based APODs in humans, they were also within 10-fold (Figure 4c). Overall, these results indicate that the BMD(C) modeling of transcriptomic data from short-term exposures can provide reasonable estimates of chronic APODs for these two different species despite several differences in study designs.

### 3.4. Gene Expression Analysis Identifies Dysregulated Molecular Pathways Previously Linked to OPP Toxicity

To investigate the possible biological significance of short-term exposures to FT, MT, and PT across both species, we combined and compared all DEGs identified in the mouse liver study and in the fathead minnow larva study after mapping to mouse orthologs. The number of unique DEGs identified in the mouse liver study was approximately 77% higher, but about 57 DEGs were consistent across the species (Figure 5a). The acetylcholinesterase (AChE) gene, a major biomarker of OPP toxicity, was among these 57 DEGs. In the mouse liver study, the AChE transcript was found to be significantly upregulated across all OPP exposures (FDR < 0.05 and maximum fold change ≥ 2), while in the fathead minnow larva study, the AChE transcript was significantly upregulated in MT (*p* = 0.009 and maximum fold change = 1.17) and PT (*p* = 0.032 and maximum fold change = 4.23) exposure groups (Figure 5b). Although the AchE transcript was not significantly upregulated in the fathead minnow FT exposure group, there was a trend towards upregulation (*p*-value = 0.18 and maximum fold change = 1.12) (Figure 5b). These results provide evidence of known molecular events in OPP toxicity. Moreover, an investigation of interaction networks showed significant regulation of genes involved in organismal stress/damage and immunological, inflammatory, and neurological biological dysfunctions (top networks scored > 50) (Appendix A). The comparison analysis across mouse liver and fathead minnow larvae also showed activation and inhibition of canonical pathways previously linked to OPP toxicity, including organismal stress and immune and neurological dysfunctions [29,30]. The top five regulated canonical pathways ranked by absolute z-score across all groups included Senescence pathway, G-protein coupled receptor signaling, Sirtuin signaling, ERK/MAPK signaling, and IL-10 signaling (Figure 5). A table of all enriched pathways for each OPP and species can be found in Appendix A. Despite some differences in the direction of regulation, there was conservation of OPP bioactivity across both species. Overall, these findings further emphasize the sensitivity in using transcriptomic response from short-term exposures to understand the OPP toxicity.

## 4. Discussion

Here, we have examined an alternative and faster approach to traditional chronic dose response assessment studies. Dose–response profiling of in vivo global gene expression changes from short-term exposures is still an underexplored area; however, it has great potential for application in risk assessment [7]. This work provides further evidence on the general consistency between short-term study-derived TPODs and chronic APODs for the three OPPs (i.e., FT, MT, and PT) across two model species. The principal finding in our studies showed that the tissue (the mouse liver study) or the whole animal (the fathead minnow larva study) transcriptome from short-term exposures (i.e., 7- and 1-day, respectively) detected chemical-induced effects at the gene set level that correspond closely to the dose levels of apical effects from chronic exposure-based studies. Our data suggest that the liver may serve as a good sentinel tissue for determining TPODs since it indicated the chronic APODs despite not necessarily being the primary OPP target (i.e., muscle and nervous tissues where acetylcholinesterase inhibition occurs). Not only were the relative potencies of the OPPs from our short-term transcriptomics-based case studies and traditional chronic exposure-based studies consistent, but the TPODs were also very similar between both species despite several differences in the experimental designs. These findings are also important because there is a critical need to be able to extrapolate molecular data collected in laboratory model organisms to apical responses (i.e., mortality, tumor formation, etc.) in a broad range of species. Acetylcholinesterase (ache), which is a major biomarker of OPP toxicity, was detected and conserved in both the mouse liver study and the fathead minnow larva study, providing confidence that we were identifying key events in OPP toxicity while also quantifying the broader concerted molecular changes occurring in both species. Understanding and linking early key molecular events to traditional apical responses can help translate high-throughput molecular-based cell assay data into toxicologically meaningful results.

Several rodent studies have shown consistency between transcriptomic BMD and TPOD values from short-term chemical exposures and concurrent or chronic BMD and APOD values. On average, the transcriptomic BMD and associated TPOD values from these studies are typically within 10-fold of the 2-year chronic-exposure apical BMD and APOD values [9,10,31,32,33]. This concordance has been previously demonstrated with industrial chemicals [8,9] and agrochemicals similar to the present study but consisting of herbicides and fungicides [31,33,34] rather than insecticides. Specifically, Bianchi and colleagues investigated the concordance of 90-day subchronic TPODs with two-year cancer bioassay PODs, using exposures to triclopyr acid, sulfoxaflor, pronamide, and fenpicoxamid. They measured transcriptomic response in rat liver or kidney for comparison to the most sensitive two-year APOD and found that it was, on average, within 2.8-fold (1.2 s.d.) [33]. In a shorter study, Bhat and colleagues compared hepatic transcriptomic TPOD responses of four tumorigenic (cyproconazole, epoxiconazole, propiconazole, and triadimefon) and one nontumorigenic (myclobutanil) conazoles to 2-year APODs for hepatocellular carcinoma and adenoma. They found that the TPODs following 30 days of exposure correctly ranked the conazoles by tumorigenic potency (cyproconazole as most potent and nontumorigenic myclobutanil as least potent), which was not possible with the associated 30-day liver-weight BMD and APOD data. Furthermore, 30-day TPODs for the four tumorigenic conazoles were generally more sensitive than the 2-year hepatocellular carcinoma and adenoma POD values and overall, within a 3.0-fold difference, on average (1.7 s.d.) [34]. In an experiment closer in duration to our mouse liver study, LaRocca and colleagues investigated the utility of using a 14-day TPOD to indicate chronic carcinogenic, reproductive, and/or endocrine APODs. They used liver and testis tissue for identifying the TPOD which was 17-fold higher than the most sensitive 2-year APOD for testis atrophy [31]. A large scale experiment of 18 well-studied chemicals used a study design quite similar to that of the present mouse liver study where Gwinn and colleagues derived transcriptomic BMDs from sentinel tissue (i.e., liver and kidney) following five-day exposures to indicate the most sensitive 2-year BMD [9]. Only BMDs were reported in the Gwinn study, and they did not use a priori knowledge of the apical effects to select the tissue for the transcriptomic analysis, yet they found that the liver- or kidney-based transcriptomic BMDs were within 10-fold of the most sensitive apical BMDs for 93% of chemicals (excluding the negative control chemicals). These studies were similar and consistent with our mouse liver study findings, which also suggested that liver may be a useful surrogate organ for determining transcriptomic BMDs and TPODs when little or no toxicity data are available for a chemical. Together, these studies suggest that TPODs from short-term exposure can be used to estimate APODs derived from chronic or subchronic exposures. 

Most of the proof-of-principle studies showing the utility of short-term transcriptomic BMDs and TPODs for indicating chronic-exposure apical BMDs and APODs have focused on rodent studies with human health applications. The mission of the U.S. EPA also encompasses protecting the environment, and an increasing number of investigators are applying BMC modeling of transcriptomics to see how well it indicates chronic PODs in ecologically relevant species such as fish. For instance, the fathead minnow, zebrafish, and rainbow trout have been used to identify TPODs after short-term exposures (ranging from 24 h to 7 days) to several chemical compounds [35,36,37,38]. These ecological studies showed that TPODs were generally conservative when compared to the subchronic and chronic apical effects for fish in several chemical exposure scenarios, suggesting that ecological TPODs can be utilized to assess the potential hazards associated with chemical exposure in adult fish. This is mostly consistent with our results because we also identified gene set TPOD values that were concordant with their respective chronic APOD values (Figure 4b). Despite these efforts, there is still a need to demonstrate additional case scenarios, validating TPOD-to-APOD correlations across diverse chemicals, species, and modes of action for this new approach to be considered as a support and/or replacement of current chronic exposure-based studies in chemical risk assessments. To our knowledge, we are the first group to report a TPOD-to-APOD correlation across mammalians and ecologically relevant species for OPPs.

Interestingly, traditional chronic apical response data for FT, MT and PT derived from the most sensitive endpoints identify FT to be the most potent OPP followed by PT and MT in several species of mammals and fish. Importantly, these results were concordant with our TPOD findings where FT was also the most transcriptionally potent OPP followed by PT and MT in the mouse liver and the fathead minnow larva studies. These observations suggest that both the 7-day and 1-day exposures in mice and fathead minnow, respectively, track the potency differences established for these three OPPs. Additionally, these comparisons made across FT, MT, and PT helped us identify potential similarities in molecular toxicity patterns of OPPs across these two species, increasing confidence in the transcriptomics-based results and contributing to the weight of evidence. We also noted that in several instances, the TPODs were more sensitive than the APODs in both species, suggesting that these data could provide conservative ecological and human health assessment values. This is consistent with the idea that a TPOD generally represents a dose/concentration below which concerted molecular changes are not expected to occur [7]. However, not all TPODs were lower than the doses/concentrations reported to elicit apical toxicity, especially for the aquatic organisms in which 50% of data points fell outside the ~10-fold difference typically seen in rodent studies (Figure 4b). This could be partly due to the duration of the fathead minnow study. The TPODs in the fathead minnow larva study may have also been a little less sensitive because the whole animal was used to obtain the transcriptomic data as opposed to only one tissue used for the mouse liver study. This may also explain the lower number of OPP-induced DEGs identified in the fathead minnow larva study. Regardless, the data suggest that the larval fathead minnow-based TPODs need to be further explored and tested against a larger set of chemicals to robustly evaluate the relationship between short-term study TPODs and traditional fish-based APODs [35]. Our case study suggests that fathead minnow larvae are a suitable organism for short-term in vivo transcriptomic testing, as it provided significant translational and functional information on the OPP effects at the molecular level. 

Organophosphates belong to a diverse group of pesticide chemicals, and they are commonly used in applications as insecticides, anthelmintics, fungicides, herbicides, etc. Exposure to OPPs elicit inhibition of acetylcholinesterase, which prevents the hydrolysis of the neurotransmitter acetylcholine, leading to hypercholinergic seizures, respiratory/cardiovascular failure, and potentially death [29]. Interestingly, our transcriptomic results showed significantly increased levels of AChE mRNA in both species. These observations are likely representative of an early-onset (i.e., acute) compensatory response to the buildup of excessive levels of acetylcholine, resulting from inhibited acetylcholinesterase enzyme activity at least in the short term. This is consistent with the current literature suggesting that AChE increases following a short-term exposure to OPPs. This has been measured in the central nervous systems of mouse and rat shortly after the OPP exposure [39,40]. This physiological compensatory response due to insecticide stress has also been observed in estuarine fish [41]. Measurement of AChE activity is a common test for biological OPP toxicity as it is expressed in the central and peripheral nervous system, muscle, and blood cells [42]. The fact that our mouse liver and fathead minnow larva-based TPODs were highly concordant with the most sensitive APOD values derived from chronic and subchronic exposure to FT, MT, and PT in humans is a remarkable finding because it shows that this new method seems consistent with the already established chronic-exposure approach to identification of APODs. 

However, we also aimed to provide an overview of FT, MT, and PT effects beyond the AchE inhibition, and another important characteristic that is worth discussing is our results regarding gene regulation and how it can affect the known mechanisms of the OPP toxicity. The enrichment analysis identified activation/inhibition of canonical pathways associated with organism damage and neurological and immune dysfunctions, which have been known to be affected by OPPs [29]. Although the pathway responses (i.e., activation and inhibition) exhibited opposing regulation directionality, these results might indicate that DEGs are sensitive to the different short-term exposures times (7 days vs. 1 day, respectively) because the regulation of DEGs is known to be time-sensitive [43]. OPP toxicity is mostly known to induce neurological effects; however, it can also disrupt organism homeostasis through stress and immune and inflammatory dysfunctions. To understand how FT, MT, and PT regulated DEGs to drive toxicity in both species, we used a network analysis. We observed significant enrichment (network score > 50; *p*-value < 0.05) of diseases and functions associated with acute immunological and inflammatory responses as well as organismal damage/stress, supporting the known information regarding OPP toxicity. Notably, the acute immune response is the first defense mechanism against exposure to persistent toxicants such as pollutants/pesticides similar to these OPPs [30]. Moreover, overstimulation of the cholinergic and glutamatergic nervous system caused by OPP exposure is usually followed by intensified generation of oxidative and inflammatory damage in many tissues [44].

Despite the bioassay-related differences, our data showed that conserved molecular changes due to short-term exposure can be used to identify TPODs, and that they correlate with APODs derived from chronic-exposure studies in mammals and fish. Furthermore, these results indicate the conservation of key events potentially important for OPP biological activity across species. While additional research is needed, these results build confidence in using short-term molecular-based assays for the characterization of toxicity and risk, thereby reducing reliance on chronic-exposure animal-based studies. 

## 5. Conclusions

The ultimate goals of this study were to (1) identify short-term study TPODs for the three OPPs (FT, MT, and PT); (2) evaluate TPOD correlation across and within species-specific APODs; and (3) further the understanding of acutely altered molecular processes associated with OPP toxicity. The TPODs were concordant across each species and generally concordant with chronic APODs, although slightly less so in the case of the 24 h fathead minnow larva assay. The upregulation of AChE transcripts was interesting and corroborates research in rodents indicating that a splice variant of AChE increases upon OPP exposure. Our results support previous reports on the mechanisms of OPP toxicity, as we found significant enrichment of genes involved in the activation and inhibition of canonical pathways that are linked to organismal stress and immune and neurological dysfunctions. Future experiments will focus on the comparative evaluation of molecular responses to other chemical classes and the corresponding apical responses across species.

## Figures and Tables

**Figure 1 toxics-11-00820-f001:**
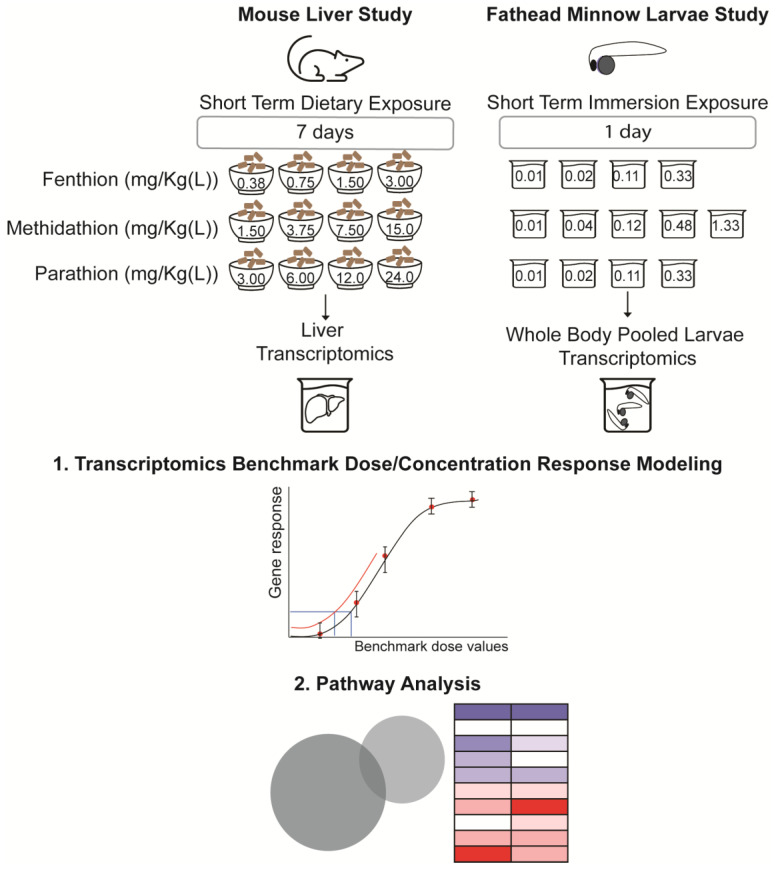
Short-term exposure plan for mouse (7 days) and fathead minnow larvae (1 day) to several dose- or concentration- levels of fenthion, methidathion, and parathion. For each treated condition, mouse liver and pooled whole-body fathead minnow larva samples were collected for targeted RNA sequencing and total RNA sequencing, respectively; transcriptomics-based benchmark dose analysis and pathway analysis.

**Figure 2 toxics-11-00820-f002:**
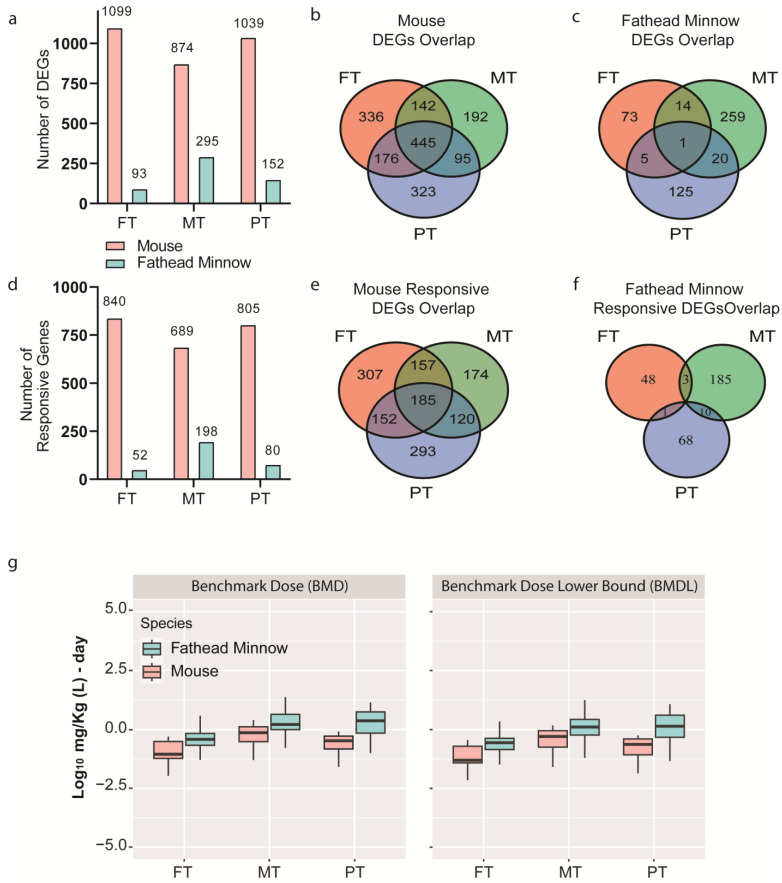
(**a**) Bar graphs represent the total number of differentially expressed genes (DEGs) for each organophosphate pesticide (fenthion (FT), methidathion (MT), and parathion (PT)) for mouse liver and fathead minnow larvae, respectively. (**b**,**c**) Venn diagram shows overlap in DEGs between FT, MT, and PT in each study. (**d**) Bar graphs represents the total number of dose- or concentration-responsive DEGs with modeled benchmark dose/concentration (BMD(C)) values. (**e**,**f**) Venn diagram shows overlap of dose- and concentration-responsive DEGs for FT, MT, and PT in each study. (**g**) Box plots represent transcriptomic BMD(C) and the lower 95%-confidence bound of BMD(C) (BMD(C)L) log-transformed median levels and ranges for FT, MT, and PT in each study. For mouse liver study, DEGs were determined by ANOVA with a false discovery rate-adjusted *p*-value < 0.05 and maximum absolute fold change across all dose levels ≥ 2. For fathead minnow larva study, DEGs were determined by ANOVA with a *p*-value < 0.05 and maximum absolute fold change across all concentration levels ≥ 2.

**Figure 3 toxics-11-00820-f003:**
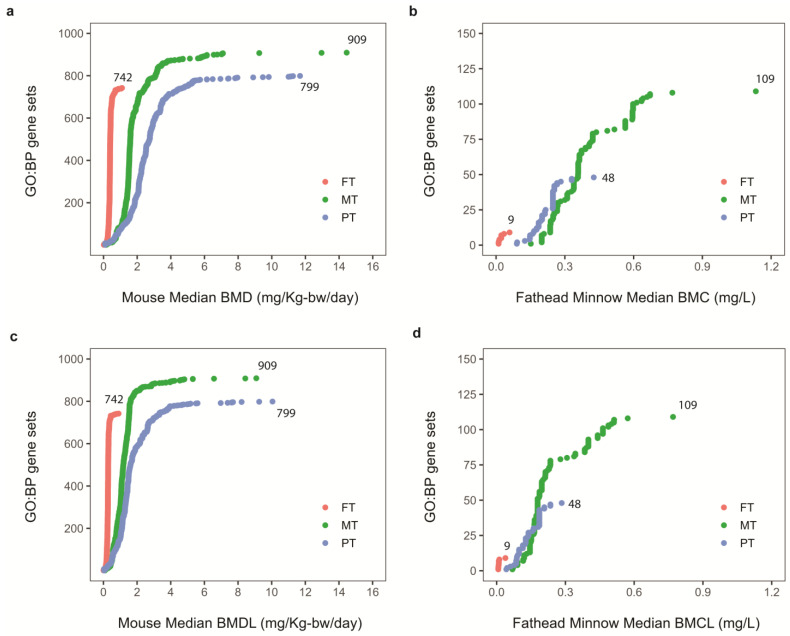
Mouse liver study and fathead minnow larva study accumulation plots showing median benchmark dose/concentration (BMD(C)) (**a**,**b**) or lower 95%-confidence bound of BMD(C) (BMD(C)L) (**c**,**d**) of significantly enriched gene-ontology biological process categories (GO:BP) for fenthion (FT) in red, methidathion (MT) in green, and parathion (PT) in blue. Significant GO:BP enrichment was determined by Fisher’s exact test two-tailed *p*-value < 0.05 and at least three genes per gene set.

**Figure 4 toxics-11-00820-f004:**
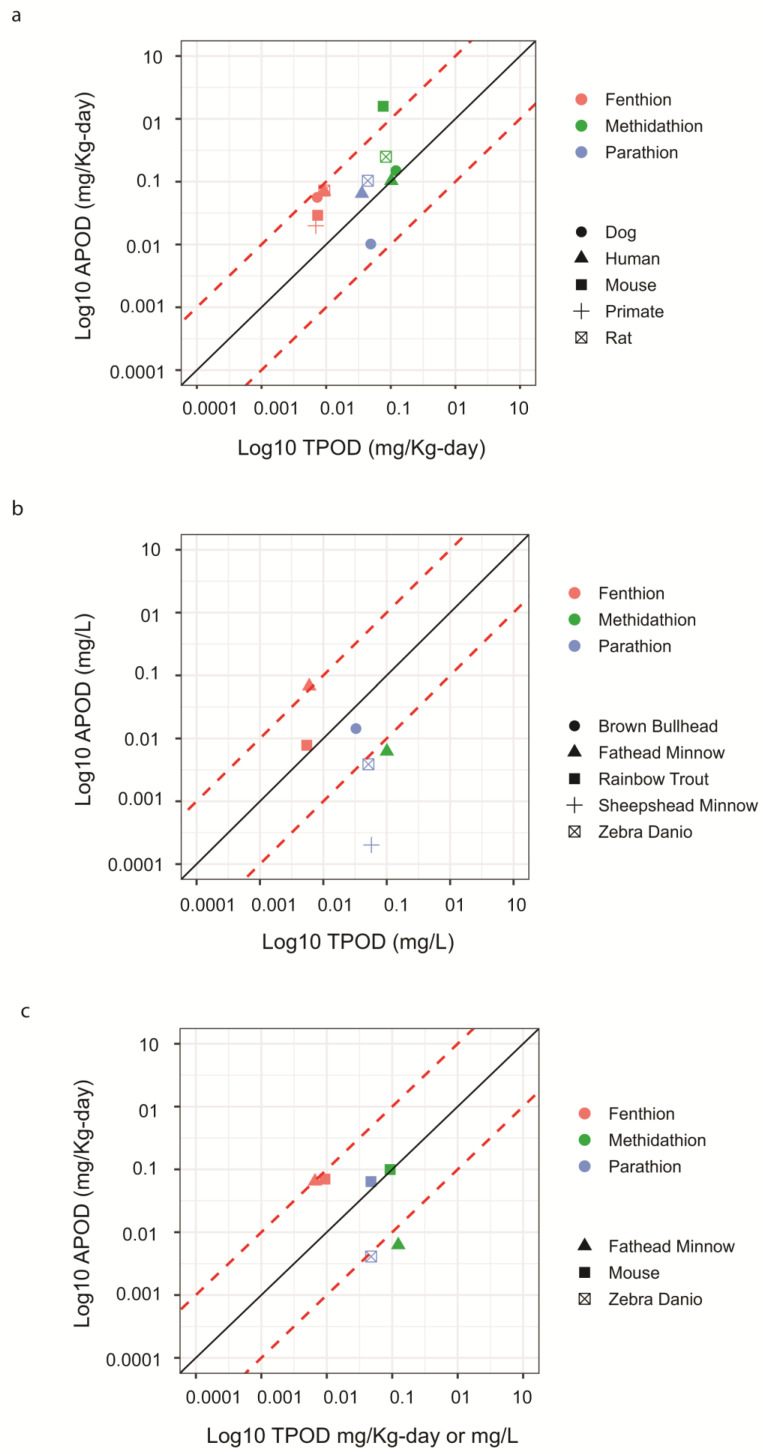
Scatter plots comparing mouse liver study and fathead minnow larva study’s transcriptomic points of departure (TPODs) identified by the benchmark dose/concentration’s 95%-confidence lower bound (BMD(C)L)) based on the median of the most sensitive significantly enriched gene-ontology biological processes (GO:BP) compared with traditional apical points of departure (APODs) identified by the no-observable-adverse-effect levels (NOAELs)) for several species of mammals (**a**), several species of fish (**b**), and humans (**c**). The solid line indicates a perfect concordance at 1. The red dashed lines represent a 10-fold difference from the solid line. Significant GO:BP enrichment was determined by Fisher’s exact test two-tailed *p*-value < 0.05 and at least three genes per gene set.

**Figure 5 toxics-11-00820-f005:**
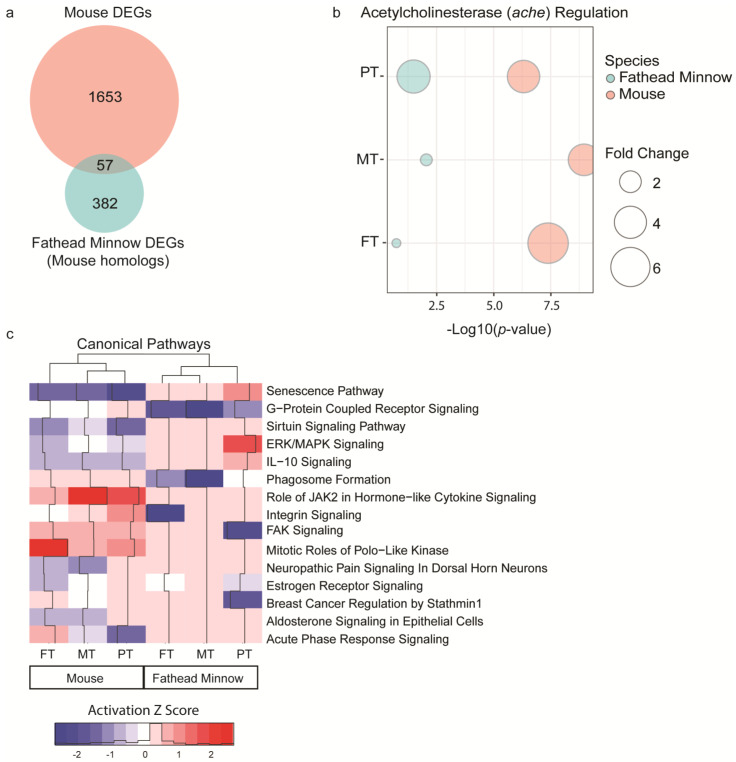
Combined differentially expressed genes (DEGs) of the mouse liver and pooled fathead minnow larva samples after exposure to fenthion (FT), methidathion (MT), and parathion (PT). (**a**) Venn diagram shows unique DEG overlap between mouse DEGs and fathead minnow DEGs converted to mouse homologs. (**b**) Bubble plot reports effects of FT, MT, and PT on acetylcholinesterase (ache) gene activity across the mouse liver study and fathead minnow larva study. The -Log10(*p*-value) was determined by ANOVA and the size of the dot is proportional to the fold change, while the pink and blue colors correspond to the two different species of interest (mouse and fathead minnow, respectively). (**c**) Comparison analysis of enriched canonical pathways (*p*-value <0.05) affected by short-term exposures to FT, MT, and PT in mouse liver study and fathead minnow larva study. The red- or blue-colored rectangles in each column indicate the z-score activities for each analysis, where red shading (positive z-scores) indicates predicted activation, and blue shading (negative z-scores) indicates predicted inhibition with ranking based on average absolute z-score across all groups. For mouse liver study, DEGs were determined by ANOVA with a false discovery rate-adjusted *p*-value < 0.05 and maximum absolute fold change ≥ 2. For fathead minnow larva study, DEGs were determined by ANOVA with a *p*-value < 0.05 and maximum absolute fold change ≥ 2 prior to mouse ortholog conversion.

**Table 1 toxics-11-00820-t001:** Organophosphate pesticides used in each study (mouse liver/fathead minnow larva studies).

Chemical	Abbreviate	CAS	Purity	Lot No.	Structure
Fenthion	FT	55-38-9	97.3%/>98%	1827400/BCBW6887	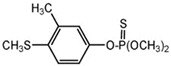
Parathion	PT	56-38-2	98.1%/98%	1757800/BCCC6538	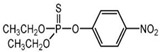
Methidathion	MT	950-37-8	99.3%/>95%	1878600/BCBX1633	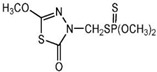

**Table 2 toxics-11-00820-t002:** Median transcriptomics-based BMD(C) and TPOD values from the most sensitive enriched gene ontology biological process (GO:BP) gene sets in mg/Kg-bw/day (mouse) or mg/L (fathead minnow).

Species	Chemicals	GO: BP	BMD(C)	BMD(C)L or TPOD	BMD(C)U
Mouse	Fenthion	Regulation of mammary gland epithelial cell proliferation	0.018	0.009	0.429
	Methidathion	Negative regulation of glucose transmembrane transport	0.164	0.093	0.306
	Parathion	Purine nucleotide transport	0.101	0.046	1.291
Fathead Minnow	Fenthion	Fatty acid metabolic process	0.011	0.007	0.120
	Methidathion	Anatomical structure homeostasis	0.151	0.115	0.217
	Parathion	Tissue development	0.090	0.046	0.682

## Data Availability

The data presented in this study, excluding sequencing FASTQ files, are openly available in Wehmas, L. (n. d.). Organophosphate pesticide manuscript dataset [Data set]. U.S. EPA Office of Research and Development (ORD). https://doi.org/10.23719/1503218 (accessed on 31 August 2023). Raw FASTQ files for the mouse liver study are openly available in the National Center for Biotechnology Information (NCBI) Gene Expression Omnibus database (accession GSE240853). Raw FASTQ files for the fathead minnow larva study are openly available in the NCBI Sequence Read Archive (submission: SUB13852035).

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
