# Peer review of "Short-Term Transcriptomic Points of Departure Are Consistent with Chronic Points of Departure for Three Organophosphate Pesticides across Mouse and Fathead Minnow"

_toxics, 2023, doi:10.3390/toxics11100820_

Round 1

Reviewer 1 Report

General comments

To my view, this paper reports the outcome of an innovative and very promising approach in basic toxicological research. It could contribute a lot to broader acceptance of so-called NAMs in future by providing evidence that there is, across species, rather good concordance between the results obtained with transcriptomics and those from more traditional testing. It is certainly worth to be published and I can confine my remarks to very few issues that might be considered by the authors to further improve the quality of this very comprehensive and convincing article.

Specific comments:

Table 1: Is there a specific reason for not displaying the chemical structure for fenthion?

Lines 122-124: Usually, the calculated dose of a pesticide is given as mg/kg bw per day. I dare to say that this is sort of general convention. It is absolutely clear to a toxicologist what you mean but a reader who is less familar with such studies might have difficulties to understand this information. This comment applies to the whole manuscript.

Lines 125/126: What is "total body weight per cage". Usually, individual body weights are determined, for all animals, and, then, a group mean body weight is calculated.

Lines 127/128: I am fine with your dose selection and, in addition, for the acutely toxic organophosphorous compounds, dose levels in studies of shorter or longer duration are mostly not that different but I find it surprising to justify the dose levels forf a 7-day study with dose selection in long-term carcinogenicity assays. One would have expected that subacute or a best subchronic studies would have been used for this purpose, instead.

Lines 138 - 142: Is this paragraph part of the description of Figure 1? I If not, I can't see where the designation of the figure ends and the normal text continues.

Subsection 2.8: Again, I have difficulties to understand why the TPODs established in a 7-day study were compared to apical PODs even from chronic studies. What I would have expected first, was a comparison between dose levels causing effects on classical toxicological parameters in the same test organisms, in your study, with those from transcriptomics. Comparison with data coming from chronic studies would be the second step. In principle, you answer my question in the discussion but from the "Material and methods" and "Results" sections, it does not become that clear.

Reviewer 2 Report

The authors provide a strong body of work that evaluated if acute exposures to organophosphate pesticides (OP) can be predictive of chronic toxicity thresholds using transcriptional point of departure (TPOD) from in vivo rat and fathead minnow embryo exposures.  Overall, the results suggest that the acute rat liver-based TPODs were more sensitive than the apical point of departures (APODs) observed in chronic mammalian exposures.  The fathead minnow embryo TPODs were less closely connected to mammalian chronic APODs, but did manage to provide analogous potency ranking among the 3 OPs tested.  So, there’s some promise to the fathead minnow assay.

Overall, the study was well conceived and executed.  The concept and the results represent very useful information for bptj the human and ecological risk communities of practice.

Fundamental Question:  Given the known neurotoxicological effects of the OPs that were investigated (principally via acetylcholinesterase inhibition), why weren’t the transcriptomics assays conducted using brain tissue instead of liver?  It seems investigation of brain would have provided a closer match to the principal toxic effects and toxicological mechanisms for this class of compounds and perhaps would have reduced some of the variance observed in the TPODs.  … Reading further, in the Discussion, I see the rationale for using liver or kidney as “sentinel” organs for screening chemical exposure.  This is acceptable, but it would still be useful to see if the brain-tissue based assay provided improved TPOD concordance with mechanism-matched APODs.  Mechanism-matched TPOD seem more defensible than just using the lowest transcriptional departure regardless of any biological connection to the toxic effect.

Specific Comments:

Materials and Methods:  The experimental methods are well described.  The experimental designs for both the rat and fathead minnow studies appear robust.  The methods used for RNAseq assays and all the downstream analyses are consistent with the state of the art and generally considered acceptable within the toxicogenomics community of practice.  Additionally, the transcriptomics benchmark dose and point of departure calculation methods represent the state of the art.

The fathead minnow assays might benefit from being longer, maybe outward to 96h hours.  This would provide the benefit of a longer exposure period and additional time for transcriptional responses to the chemical(s) of concern, while still operating within an embryo exposure.  The economy of the assay wouldn’t be lost by adding an extra 3 days, especially when the goal is to predict POD for 2-year rodent studies.
